# 2,5-Diketopiperazine Derivatives as Potential Anti-Influenza (H5N2) Agents: Synthesis, Biological Evaluation, and Molecular Docking Study

**DOI:** 10.3390/molecules27134200

**Published:** 2022-06-29

**Authors:** Chanakan Winyakul, Weerachai Phutdhawong, Poomipat Tamdee, Jitnapa Sirirak, Thongchai Taechowisan, Waya S. Phutdhawong

**Affiliations:** 1Department of Chemistry, Faculty of Science, Silpakorn University, Nakorn Pathom 73000, Thailand; chanakanwnkn@gmail.com (C.W.); tamdee_p@silpakorn.edu (P.T.); jitnapasirirak@gmail.com (J.S.); 2Department of Chemistry, Faculty of Liberal Arts and Science, Kasetsart University, Kamphaengsaen Campus, Nakorn Pathom 73140, Thailand; phutdhawong@gmail.com; 3Department of Microbiology, Faculty of Science, Silpakorn University, Nakorn Pathom 73000, Thailand; tewson84@hotmail.com

**Keywords:** 2,5-Diketopiperazines, antiviral activity, influenza virus, molecular docking, Lansai C, Lansai D

## Abstract

2,5-Diketopiperazine derivatives, consisting of benzylidene and alkylidene substituents at 3 and 6 positions, have been considered as a core structure for their antiviral activities. Herein, the novel *N*-substituted 2,5-Diketopiperazine derivatives were successfully prepared and their antiviral activities against influenza virus were evaluated by monitoring viral propagation in embryonated chicken eggs. It was found that (3*Z*,6*Z*)-3-benzylidene-6-(2-methyl propylidene)-4-substituted-2,5-Diketopiperazines (**13b–d**), (3*Z*,6*E*)-3-benzylidene-6-(2-methylpropyli dene)-1-(1-ethyl pyrrolidine)-2,5-Diketopiperazine (**14c**), and Lansai-C exhibited negative results in influenza virus propagation at a concentration of 25 µg/mL. Additionally, molecular docking study revealed that **13b**–**d** and **14c** bound in 430-cavity of neuraminidase from H5N2 avian influenza virus and the synthesized derivatives also strongly interacted with the key amino acid residues, including Arg371, Pro326, Ile427, and Thr439.

## 1. Introduction

2,5-Diketopiperazine (2,5-DKP) is a six-membered cyclic dipeptide which is often found alone or embedded as a part of a compound in a variety of natural products from microorganisms, plants, and animals. 2,5-DKP can bind to a wide variety of receptors due to its unique structure, which includes a rigidity, chirality, and a variety of side chains. As a result, 2,5-DKP scaffolds are widely used for drug discovery [1,2,3,4,5]. Additionally, 2,5-DKP compounds exhibit many bioactivities, which include antiviral activity [6].

For example, Aplaviroc exhibited high binding affinity to chemokine co-receptor 5, and was developed for the treatment of patients with human immunodeficiency virus type 1 (HIV-1) [1]. Eutypellazine A–L isolated from fungus *Eutypella* sp. MCCC 3A00281 could inhibit the replication of HIV-1 with low toxicity (CC_50_ > 100). Eutypellazine E showed an inhibitory effect at IC_50_ 3.2 + 0.4 µM. The structure–activity relationship study of these 2,5-DKP compounds showed that the thiomethyl group at C-2/C-2′ and the double bond at C-6′/C-7′ in Eutypellazine E remarkably enhanced the activity compared with analogs [7]. In addition, Rubrumlines D and Neoechinulin B isolated from the fungus *Eurotium rubrum* exhibited antiviral activity against the influenza A/WSN/33 virus with inhibitory rates of 52.64% and 70.48% (IC_50_ 126.0 and 27.4 and CC_50_ > 200), respectively. The presence of a double bond at the ∆^12,15^ unit of Rubrumlines D remarkably enhanced the antiviral activity. On the contrary, Neoechinulin B, containing the absence of isoprenyl or an oxygenated isoprenyl group at the indole ring (Figure 1), showed the highest antiviral activity [8]. Additionally, Wang and co-workers reported that Albonoursin and its derivative (R=OH, (3*Z*,6*Z*)-3-(4-hydroxybenzylidene)-6-isobutylidenepiperazine-2,5-dione) showed influenza virus activity against H1N1 with IC_50_ at 41.5 ± 4.5 and 6.8 ± 1.5 µM, respectively [9].

The other derivatives of 2,5-DKP are Lansai C (LS-C) and Lansai D (LS-D), which can be obtained from *Streptomyces* sp. SUC1 isolated from the aerial roots of *Ficus benjamina* in our campus [10,11]. LS-C and LS-D exhibit an anti-inflammatory effect on RAW 264.7 cells [12,13]. The structures of LS-C and LS-D contain benzylidene and alkylidene substituents at 3 and 6 positions, which are similar to that of Albonoursin, except their double bond configurations. Thus, LS-C and LS-D and their antiviral activities are worth investigation. Moreover, the modification of the *N*-substituents of 2,5-DKPs of Albonoursin is interested, since the functionalized chains of 2,5-DKPs and their orientations can affect the binding affinity of 2,5-DKPs to the receptors and the biological activities of 2,5-DKPs compounds [14].

In this research, the novel 2,5-DKP derivatives, which were designed by emulating Albonoursin, LS-C, and LS-D scaffolds with the addition of substitution group at the position of nitrogen atoms of the 2,5-DKP ring by *N*-alkylation reaction, were successfully prepared. The preliminary antiviral activity test against influenza virus (H5N2) of our 2,5-DKP derivatives, LS-C, and LS-D was then performed by hemagglutination assay. In addition, a molecular docking study was conducted to investigate how the derivatives interact with amino acids in the binding pocket of neuraminidase from H5N2 avian influenza virus (H5N2, PDB ID: 5HUK). Our work could contribute to and benefit the design and development of novel 2,5-DKP derivatives as a potential antiviral drug.

## 2. Results and Discussion

### 2.1. Synthesis

The steps to synthesize tetrasubstituted 2,5-DKP derivative **5** were as follows. *N*-Boc-Leucine **1** and Phenylalanine methyl ester **2** were reacted to obtain dipeptide **3**. Then, the deprotection of the Boc group, followed by ring formation of the dipeptide **3**, was conducted to obtain 2,5-DKP derivative **5**. Finally, *N*-substitution of the 2,5-DKP derivative **5** led to the formation of tetrasubstituted 2,5-DKP derivatives **6**–**9** (Figure 1) in fair yields.

The 2,5-DKP ring **10** was constructed with moderated yield (56%) in accordance with previous literature to obtain benzylidene and alkylidene substituents at 3 and 6 positions of the 2,5-DKP core structure [15]. *N*-acylation of **10**, by refluxing in acetic anhydride, provided 1,4-Diacetyl-2,5-diketopiperazine **11** in fair yield (Figure 2).

1,4-Diacetyl-2,5-diketopiperazine **11** underwent aldol addition–acetyl migration–elimination cascade with Cs_2_CO_3_, alkyl halide, and various aldehydes [16] to provide highly (*Z*)-stereoselective products **12a**–**e** (Figure 2). This reaction was supported by the Zimmerman–Traxler model, which was reported by Balducci and co-workers [17]. In the case of *N*-benzyl derivative **12g**, the cascade reaction using benzyl chloride failed to provide the desired product, probably because of the steric hindrance with the benzylidene substituent. Thus, the benzylidene derivative **12f** was further benzylated to provide compound **12g**. Then, the next aldol condensation was accomplished to provide trisubstituted 2,5-DKP derivatives (**13a**–**g**). However, **13a**–**g** have low solubility in chromatographic solvent, leading to low product yields [18].

Compound **13f** was a natural product from Streptomyces noursei, which was known as Albonoursin [19]. *N*-alkylation of compound **13f** with NaH, benzyl chloride, and 1-chloro-3-methyl-2-butene provided disubstituted products **14a** and **14b**. However, only monosubstituted product, **14c**, was obtained when using the bulky 1-ethyl pyrrolidine chloride. The propylidene was then converted to *E*-configuration, probably because of the steric hindrance (Figure 3).

The configuration of the double bond was confirmed by NMR spectroscopy. In this research, each proton NMR spectrum of the C=CH double bond in 6-(2-methylpropylidene) was observed at 6.01–6.13 ppm, which might be the indication of (6*Z*)-configuration similar to the observation of Fairhurst and co-workers’ work, in which the chemical shifts of proton of *Z*- and *E*-isomers at the position of an alkylidene double bond (C=CH) in complex tert-butyl-3-(2-methylpropylidene)-2,5-dioxopiperazine-1-carboxylate were 6.14 and 5.42 ppm, respectively [20]. The NOEDIFF spectra of compound **12e** showed that the proton on the alkylidene double bond (C=CH) only enhanced the proton of isopropyl moiety. Thus, the alkylidene double bond of compound **12e** was characterized as (*Z*)-configuration. Another NOEDIFF analysis of compound **13d** showed that 4-NH and H-11 enhanced Ar-H of benzylidene moiety and 1-*N*-CH_3_, respectively. Thus, compound **13d** was characterized as (3*Z*,6*Z*)-3-benzylidene-6-(2-methylpropylidene)-1-methyl-2,5-diketopiperazine (Figure 2). Other results showed steric hindrance between the proton of the aromatic ring and the carbonyl and steric repulsion between hydrogen atoms in benzylidene and the *N*-alkylated group, which could be caused by (3*Z*)-selectivity. Further, NOESY analysis was used to confirm the stereochemistry of (6*E*)-configuration in compound **14c**. NOE enhancement from H-7 to H-18, H-19, H-21, H-22, H-23, and H-24 was observed, whereas NOE enhancement of H-8 was not found. This result indicated (6*E*)-configuration (Figure 2).

### 2.2. Virus Propagation Inhibition Assay

The efficacy of LS-C, LS-D, and 2,5-DKP derivatives to influenza virus (H5N2) propagation inhibition was evaluated at various concentrations in embryonated chicken eggs. A hemagglutination test was performed to estimate virus propagation. The summary results are displayed in Table 1. The negative control (PBS) showed the ratio of the last dilution at 1:3072, indicating complete agglutination. On the other hand, 1-adamantanamine hydrochloride and oseltamivir carboxylate were used as the positive controls. The results show that they exhibited the virus inhibition at 12.5 µg/mL while all of our tested compounds could not inhibit the virus propagation at a concentration of 12.5 µg/mL.

LS-C showed the virus inhibition at concentration of 25 µg/mL, whereas LS-D could not inhibit virus propagation even at a high concentration (100 µg/mL). The result indicated that N-OH group of LS-C were essential for the antiviral activity. Moreover, compound **13d**, containing opposite double bond configuration to LS-D, showed an inhibition against influenza virus (H5N2) propagation at a concentration of 25 µg/mL, suggesting that the double bond in *Z*-configuration might enhance the activity of the compound. Additionally, either the unprotected nitrogen or protected nitrogen with alkyl substituents such as allyl and 3-methyl-2-butene adjacent to (*Z*)-benzylidene inhibited virus propagation at a concentration of 25 µg/mL, as was observed in compounds **13b**–**d** and compound **14c**. In contrast, the benzyl substituent **13g** had a different impact on the antiviral activity, thus there should not be any substitution on nitrogen adjacent to isopropylidene, for the isopropylidene is in *Z*-configuration in order to maintain the antiviral activity. The presence of double bonds was also important for the antiviral activity in compounds **6**–**9**.

In addition, the cytotoxicity of LS-C, LS-D, **13b–d**, and **14c**, which exhibited the virus inhibition at a concentration of 25 µg/mL, were investigated against Rhesus monkey kidney epithelial cells (LLC-MK2 cell lines). It was found that their cytotoxicity activities (IC_50_) against LLC-MK2 were 287.65 to 507.84 µg/mL, indicating that cytotoxicities of these compounds were negligible to the normal cells.

### 2.3. Molecular Docking Study

The molecular docking is a computer-aided procedure that generates ligand with different orientations and conformation and predicts the best match between ligand and protein target, where the lower the binding energy of a complex indicates that the complex is more stable [21]. Herein, molecular docking approach is employed not only to visualize how our compounds interact with neuraminidase from H5N2 avian influenza virus, which was the well-known drug target to prevent the spread of influenza infection [22], but also to gain valuable guidance at a molecular level for the development of new 2,5-DKP derivatives as the antiviral drugs.

According to the virus propagation inhibition assay results, the following compounds: LS-C and compounds **7**, **9**, **13a**–**13d**, and **14c**, which serve as virus propagation inhibitor, were selected. Molecular docking study was performed using iGEMDOCK v.2.1. [23] and the images were prepared using BIOVIA Discovery Studio Visualizer [24]. Additionally, an antiviral medication used to treat and prevent influenza viruses, including oseltamivir carboxylate [25,26], zanamivir [27], and peramivir [28], were docked into neuraminidase from H5N2 avian influenza virus, and their docking results were compared with our potential antiviral drugs (LS-C and compounds **7**, **9**, **13a**–**13d**, and **14c**). The initial structures of oseltamivir carboxylate, zanamivir, and peramivir were taken from crystal structures of neuraminidase-inhibitor complexes PDB ID: 2HU4, 3CKZ, and 3K39, respectively.

The comparison of binding positions of our selected compounds with virus propagation inhibitor (LS-C and compounds **7**, **9**, **13a**–**13d**, and **14c**) in the active site of H5N2 was displayed in Figure 3a. The binding energies, amino acid interaction, along with hydrogen bond length of our compounds bound in the active site of receptors, were also demonstrated in Table 2. As can be seen in Figure 3b and Table 2, oseltamivir carboxylate, zanamivir, and peramivir fit and interact with amino acids in the sialic acid cavity of H5N2 in similar manner to the observation in the crystal structures of neuraminidase-inhibitor complexes [29,30,31], which validates our docking method. Moreover, compound **7** binds in the front of sialic acid cavity of H5N2 and forms hydrogen bond with ASN249 due to its large structure containing three benzyl moieties.

On the other hand, the LS-C and compound **9, 13a**–**13d**, and **14c**, are located in a hydrophobic 430-cavity of H5N2, as shown in Figure 3c. The interaction between these compounds and key amino acid residue, PRO326, ILE427, and THR439, are also observed. Moreover, their carbonyl groups on 2,5-DKP scaffold strongly interact with ARG371, which is one of the arginine triad residues (ARG118-ARG292-ARG371), indicating the important role of 2,5-DKP scaffold for H5N2-binding and H5N2 inhibition. Hydrogen bond interactions between some of our compounds (**13c**, **13d**, and **14c**) are represented in Figure 4. In comparison to binding energy, our compounds have higher binding energy (−102.25 to −77.18 kcal/mol) than zanamivir and peramivir (−107.07 and −115.88 kcal/mol, respectively), while only compound **13d** and **14c** exhibit lower binding energy (−101.86 and −102.25 kcal/mol, respectively) than oseltamivir carboxylate (−93.75 kcal/mol). This result suggests that the hydrophilic substitute groups on 2,5-DKP scaffold are required for increasing the binding efficiency and development of the novel antiviral drugs.

## 3. Materials and Methods

### 3.1. General Experimental Procedures

All reactions sensitive to air or moisture were carried out under anhydrous conditions, unless otherwise stated. Solvents and reagents were used without further purification. The reagents were purchased from Sigma–Aldrich (Darmstadt, Germany), Tokyo Chemical Industry (Tokyo, Japan), and Fluka Chemical (Buchs, Switzerland) Companies. ^1^H- and ^13^C-NMR spectra were measured in CDCl_3_ or DMSO-d_6_ by a Bruker Avance 300 spectrometer (Bruker, Massachusetts, USA; 300 MHz for ^1^H, 75 MHz for ^13^C). Melting points were measured by using a Stuart Scientific SMP 2 melting point apparatus (Cole-Parmer Ltd., Staffordshire, UK). Mass spectra were measured by using micrOTOF (Bruker, Billerica, MA, USA). The reactions were monitored by thin-layer chromatography (TLC) and by using an aluminum sheet pre-coated with silica gel 60 F_254_ (Merck, Darmstadt, Germany). Column chromatography was conducted on Merck Kieselgel 60 (Merck, Darmstadt, Germany). (*N*-tert-butoxycarbonyl)-l-leucine **1** and l-Phenylalanine methyl ester hydrochloride **2** were prepared in accordance with the previous literatures [32,33]. Lansai A, Lansai B, LS-C, and LS-D were isolated, purified, and identified as previously described [10,11]. The ^1^H-NMR, ^13^C-NMR, HRMS spectra of the synthesized compounds were recorded, as shown in the Appendix A.

### 3.2. Synthesis of 2,5-DKP Derivatives

#### 3.2.1. Methyl (*N*-tert-butoxycarbonyl)-l-leucyl-l-phenylalaninate (**3**)

(*N*-tert-butoxycarbonyl)-l-leucine (**1**) (0.11 g, 0.40 mmol) was dissolved in dry THF (3.0 mL) under argon atmosphere and the reaction was kept at 0 °C in an ice bath. The solution was treated with HATU (0.25 g, 0.70 mmol) and DIPEA (0.11 mL, 0.60 mmol) and allowed to stir at 0 °C for 10 min. Then, l-phenylalanine methyl ester hydrochloride (**2**) (0.09 g, 0.50 mmol) was added, and the reaction was stirred at room temperature overnight. After completing the reaction, the solvent was removed under vacuum, and the residue was dissolved with EtOAc. The EtOAc solution was extracted with 2N HCl (2 × 20.0 mL) and saturated NaHCO_3_ (2 × 20.0 mL). The organic phase was washed with brine (2 × 20.0 mL), dried with anhydrous Na_2_SO_4_, and evaporated under reduced pressure to provide the crude product. The crude product was purified by column chromatography using hexane:EtOAc (4:1) to obtain a colorless oil ((0.11 g, 84%yield); ^1^H-NMR (300 MHz, CDCl_3_) δ 7.30–7.19 (m, 3H), 7.12–7.10 (m, 2H), 6.72 (m, 1H), 5.03 (s, 1H), 4.84 (dd, *J* = 13.5, 6 Hz, 1H), 4.10 (m, 1H), 3.68 (s, 1H), 3.14 (dd, *J* = 13.8, 5.7 Hz, 1H), 3.06 (dd, *J* = 13.8, 6.3 Hz, 1H), 1.69–1.25 (m, 2H), 1.49–1.39 (m, 1H), 1.43 (s, 3H), 0.91 (d, *J* = 6.3 Hz, 3H), 0.90 (d, *J* = 6 Hz, 3H); ^13^C-NMR (300 MHz, CDCl_3_) δ 172.3, 171.7, 155.5, 135.8, 129.3, 128.5, 127.0, 80.0, 53.1, 52.2, 52.1, 41.2, 37.9, 28.2, 24.6, 22.8, 21.9.

#### 3.2.2. Methyl-l-leucyl-l-phenylalaninate (**4**)

Methyl (tert-butoxycarbonyl)-l-leucyl-l-phenylalaninate **3** (0.54 g, 1.40 mmol) was dissolved in DCM (5.0 mL). Trifluoroacetic acid (0.2 mL, 2.60 mmol) was added to the solution at 0 °C. The reaction mixture was stirred at room temperature for 2 h. Then, the excess reagent and solvent were removed under reduced pressure. The resulting oil was treated follows: neutralized by saturated NaHCO_3_, extracted with DCM (2 × 30.0 mL), dried with anhydrous Na_2_SO_4_, and evaporated under reduced pressure. The crude product was purified by column chromatography using 5% MeOH in DCM to obtain a colorless oil (0.28 g, 68% yield); ^1^H-NMR (300 MHz, CDCl_3_) δ 7.72 (d, *J* = 8.1 Hz, 1H), 7.30–7.20 (m, 3H), 7.13–7.11 (m, 2H), 4.83 (dd, *J* = 13.8, 7.5 Hz, 1H), 3.70 (s, 3H), 3.36 (m, 1H), 3.16 (dd, *J* = 13.8, 6.0 Hz, 1H), 3.05 (dd, *J* = 13.6, 6.9 Hz, 1H), 1.70–1.55 (m, 1H), 1.60–1.52 (m, 1H), 1.25–1.17 (m, 1H), 0.92 (d, *J* = 6.3, Hz, 3H), 0.88 (d, 6.3 Hz, 3H); ^13^C-NMR (300 MHz, CDCl_3_) δ 175.7, 172.3, 136.1, 129.2, 128.5, 127.0, 53.3, 52.8, 52.3, 43.9, 37.9, 24.7, 23.2, 21.4.

#### 3.2.3. (3*S*,6*S*)-3-Benzyl-6-isobutyl-2,5-diketopiperazine (**5**)

Methyl-l-leucyl-l-phenylalaninate (**4**) (0.31 g, 1.10 mmol) was dissolved in sec-buthanol:toluene (1:4, 12.0 mL). The solution was refluxed at 90 °C for 4 h. Then, the reaction was worked up by removing solvent under vacuum. Finally, the residue was triturated with MTBE to obtain a white solid (0.12 g, 46% yield); m.p. 258 °C–259 °C (Lit [34] 263 °C–264 °C); ^1^H-NMR (300 MHz, CDCl_3_) δ 7.34–7.30 (m, 3H), 7.25–7.22 (m, 2H), 6.21 (s, 1H) 6.13 (s, 1H), 4.26–4.22 (m, 1H), 3.36–3.32 (m, 1H), 3.31 (dd, *J* = 13.9, 3.8 Hz, 1H), 3.05 (dd, *J* = 13.8, 7.8 Hz, 1H), 1.76–1.71 (m, 1H), 1.69–1.64 (m, 1H), 1.57–1.50 (m, 1H), 0.92 (d, *J* = 6.2 Hz, 3H, 0.83 (d, *J* = 6.2 Hz, 3H); ^13^C-NMR (300 MHz, CDCl_3_) δ 168.7, 167.7, 135.0, 129.7, 129.0, 127.6, 56.1, 52.7, 41.8, 39.8, 24.1, 23.1, 21.0.

#### 3.2.4. (3*S*,6*S*)-3-Benzyl-6-isobutyl-1,4-dimethyl-2,5-diketopiperazine (**6**)

(3*S*,6*S*)-3-Benzyl-6-isobutyl-2,5-diketopiperazine (**5**) (0.08 g, 0.30 mmol) was dissolved in DMF (3.0 mL) under argon atmosphere. Cs_2_CO_3_ (0.32 g, 0.90 mmol). Next, methyl iodide (0.06 mL, 0.90 mmol) was added to the solution. The reaction mixture was then stirred at 0 °C for 2 h. After that, the reaction was quenched with brine and extracted with EtOAc (2 × 20.0 mL). The organic layer was washed with brine (3 × 20.0 mL) and dried with anhydrous Na_2_SO_4_. Finally, the crude product was obtained by removing EtOAc solvent under reduced pressure and purified by preparative TLC using methanol in DCM (5%) as a mobile phase to obtain a colorless oil ((0.01 g, 16% yield); ^1^H-NMR (300 MHz, CDCl_3_) δ 7.32–7.24 (m, 3H), 7.12–7.09 (m, 2H), 4.18 (t, *J* = 4.6 Hz, 1H), 3.60 (dd, *J* = 9.2, 4.2 Hz, 1H), 3.30 (dd, *J* = 13.9, 4.8 Hz, 1H), 3.16 (dd, *J* = 13.9, 4.5 Hz, 1H), 2.95 (s, 3H), 2.86 (S, 3H), 1.76–1.63 (m, 1H), 0.82 (d, *J* = 6.5 Hz, 3H), 0.70 (d, *J* = 6.6 Hz, 3H), 0.69–0.60 (m, 1H), 0.29–0.19 (m, 1H); ^13^C-NMR (300 MHz, CDCl_3_) δ 166.5, 165.1, 153.6, 130.0, 128.8, 127.6, 63.9, 60.0, 42.4, 37.6, 32.8, 32.5, 25.1, 22.7, 21.4; HRMS [ESI]^+^ calculated for C_17_H_24_N_2_O_2:_ 311.1730 [M + Na]^+^; found: 311.1733).

#### 3.2.5. General Procedure for the Synthesis of (3*S*,6*S*)3-Benzyl-1,4-disubstituted-6-isobutyl-2,5-diketopiperazine (**7**–**9**)

(3*S*,6*S*)-3-Benzyl-6-isobutyl-2,5-diketopiperazine **5** (0.40 mmol) was dissolved in DMF (3.0 mL) under argon atmosphere. The solution was cooled to 0 °C followed by addition of NaH (2.2 eq), and the mixture was stirred at 0 °C for 15 min. Next, alkyl halide/aryl halide (2.0 eq) and TBAI (0.2 eq) were added to the solution. The reaction was stirred at room temperature for 4 h. The reaction was then quenched with saturated NH_4_Cl and extracted with EtOAc (3 × 20.0 mL). After that, the organic layer was washed by water (3 × 20.0 mL) then dried with anhydrous Na_2_SO_4_, followed by evaporation under reduced pressure. Finally, the crude product was purified by preparative TLC (silica gel, EtOAc: hexane, 1:4).

##### (3*S*,6*S*)-1,3,4-Tribenzyl-6-isobutyl-2,5-diketopiperazine (**7**)

Following the general procedure, the product was obtained as a colorless oil (0.07 g, 34% yield); ^1^H-NMR (300 MHz, CDCl_3_) δ 7.36–7.27 (m, 9H), 7.18–7.12 (m, 4H), 7.10–7.07 (m, 2H), 5.37 (d, *J* = 15.4 Hz, 1H), 5.32 (d, *J* = 15.4 Hz, 1H), 4.23 (dt, *J* = 5.2, 0.9 Hz, 1H), 3.78 (d, *J* = 15 Hz, 1H), 3.74 (dd, *J* = 10.2, 3.9 Hz, 1H), 3.57 (d, *J* = 15 Hz, 1H), 3.31 (dd, *J* = 14.0, 4.7 Hz, 1H), 3.23 (dd, *J* = 14.1, 6.0 Hz, 1H), 1.79–1.74 (m, 1H), 1.02–0.93 (m, 1H), 0.82 (d, *J* = 6.3 Hz, 3H), 0.76 (d, *J* = 6.6 Hz, 3H), 0.69–0.60 (m, 1H); ^13^C-NMR (300 MHz, CDCl_3_) δ 166.9, 165.9, 136.0, 135.7, 135.6, 129.9, 128.9, 128.8, 128.1, 128.0, 127.9, 127.6, 60.6, 56.9, 47.1, 47.0, 42.7, 38.1, 25.2, 22.9, 21.4; HRMS [ESI]^+^ calculated for C_29_H_32_N_2_O_2:_ 463.2356 [M + Na]^+^; found: 463.2362).

##### (3*S*,6*S*)-3-Benzyl-6-isobutyl-1,4-bis(3-methylbut-2-en-1-yl)-2,5-diketopiperazine (**8**)

Following the general procedure, the product was obtained as a colorless oil (0.07 g, 47% yield); ^1^H-NMR (300 MHz, CDCl_3_) δ 7.32–7.20 (m, 3H), 7.15–7.12 (m, 2H), 5.12–5.09 (m, 1H), 5.08–5.04 (m, 1H), 4.58 (dd, *J* = 14.7, 5.7 Hz, 1H), 4.40 (dd, *J* = 14.9, 5.6 Hz, 1H), 4.24 (T, *J* = 4.9 Hz, 1H), 3.72 (dd, *J* = 10.1, 3.7 Hz, 1H), 3.46 (dd, *J* = 14.8, 8.2 Hz, 1H), 3.37 (dd, *J* = 14.8, 8.5 Hz, 1H), 3.25 (dd, *J* = 14.1, 5.1 Hz, 1H), 3.14 (dd, *J* = 14.1, 4.8 Hz, 1H), 1.75–1.69 (m, 1H), 1.72 (s, 3H), 1.71 (s, 3H), 1.65 (s, 3H), 1.63 (s, 3H), 0.83–0.75 (m, 1H), 0.82 (d, *J* = 6.5 Hz, 3H), 0.70 (d, *J* = 6.6 Hz, 3H); ^13^C-NMR (300 MHz, CDCl_3_) δ 166.4, 165.3, 137.8, 137.1, 136.0, 130.0, 128.7, 127.4, 118.5, 118.1, 60.3, 56.9, 42.2, 41.8, 41.5, 37.8, 25.8, 25.7, 24.9, 23.0, 21.2, 18.0, 17.9 HRMS [ESI]^+^ calculated for C_25_H_36_N_2_O_2:_ 419.2669 [M + Na]^+^; found: 419.2673).

##### (3*S*,6*S*)-1,4-Diallyl-3-benzyl-6-isobutyl-2,5-diketopiperazine (**9**)

Following the general procedure, the product was obtained as a colorless oil (0.02 g, 15% yield); ^1^H-NMR (300 MHz, CDCl_3_) δ 7.33–7.25 (m, 3H), 7.15–7.12 (m, 2H), 5.78–5.71 (m, 1H), 5.70–5.64 (m, 1H), 5.23 (ddd, *J* = 10.2, 4.2, 0.9 Hz, 2H), 5.14 (ddd, *J* = 17.1, 9.6, 0.9 Hz, 2H), 4.68 (tdd, *J* = 15.0, 4.8, 1.5 Hz, 1H), 4.54, (tdd, *J* = 15.0, 4.8, 1.5 Hz, 1H), 4.28 (t, *J* = 5.0 Hz, 1H), 3.75 (dd, *J* = 9.8, 3.8 Hz, 1H), 3.36 (dd, *J* = 15.0, 7.5 Hz, 1H), 3.27–3.15 (m, 1H), 3.22–3.18 (m, 2H), 1.80–1.64 (m, 1H), 0.90–0.81 (m, 1H), 0.84 (d, *J* = 6.6 Hz, 3H), 0.73 (d, *J* = 6.6 Hz, 3H), 0.53–0.40 (m, 1H); ^13^C-NMR (300 MHz, CDCl_3_) δ 166.5, 165.3, 135.8, 131.8, 131.4, 130.0, 128.8, 127.5, 119.1, 118.8, 60.5, 57.0, 46.6, 42.6, 37.9, 25.2, 22.9, 21.3; HRMS [ESI]^+^ calculated for C_21_H_28_N_2_O_2:_ 341.2224 [M + H]^+^; found: 341.2224).

#### 3.2.6. 2,5-Diketopiperazine (**10**)

Methyl (tert-butoxycarbonyl) glycyl glycinate [35] (**12**) (0.54 g, 1.40 mmol) was dissolved in DCM (5.0 mL). Trifluoro acetic acid (0.2 mL, 2.60 mmol) was added to the solution at 0 °C. The reaction mixture was left at room temperature for 2 h. Next, the excess reagent and solvent were removed under reduced pressure. The resulting oil was then neutralized by saturated NaHCO_3_, followed by extraction by DCM (2 × 30.0 mL), drying with anhydrous Na_2_SO_4_, and evaporation under reduced pressure to obtain a colorless oil. After that, the crude product was dissolved in sec-buthanol:toluene (1:4, 12.0 mL). The prepared solution was refluxed at 90 °C for 4 h. The reaction was then worked up by removing solvent under vacuum. Finally, the residue was triturated by MTBE to obtain a white solid (0.25 g, 56% yield); m.p. 309 °C–310 °C (Lit [36] 311 °C–312 °C); ^1^H-NMR (300 MHz, CDCl_3_) δ 4.04 (s, 2H); ^13^C-NMR (300 MHz, CDCl_3_) δ 168.6, 43.9.

#### 3.2.7. 1,4-Diacetyl-2,5-diketopiperazine (**11**)

2,5-Diketopiperazine (**10**) (0.51 g, 4.5 mmol) was suspended in acetic anhydride (6.0 mL) followed by the addition of two drops of concentrated H_2_SO_4_. The reaction mixture was refluxed at 100 °C for 2.5 h. The reaction provided the red solution, which was diluted with EtOAc and filtered through celite. The filtrate was concentrated to obtain the crude product as a yellow oil. The yellow oil was recrystallized by using excess isopropanol and was then stored in a fridge to obtain the precipitated pure product as a colorless needle crystal. The crystal was filtered and was then washed by cold isopropanol (0.19 g, 21% yield); m.p. 99–100 °C (Lit [37] 102–103 °C); ^1^H-NMR (300 MHz, CDCl_3_) δ 4.60 (s, 2H), 2.59 (S, 3H); ^13^C-NMR (300 MHz, CDCl_3_) δ 170.7, 165.8, 47.1, 26.7.

#### 3.2.8. General Procedure for the Synthesis of (3*Z*)-4-Acetyl-3-benzylidene-1-substituted-2,5-diketopiperazine (**12a**–**e**)

A 10 mL round bottom flask, which contained 1,4-diacetyl poperazine-2,5-dione (**11**) (0.10 g, 0.54 mmol, 1.0 eq), aldehyde (1.40 mmol, 2.5 eq), alkyl halide (1.40 mmol, 2.5 eq), Cs_2_CO_3_ (1.40 mmol, 2.5 eq), and dry DMF (4.4 mL), was filled with argon atmosphere. The reaction mixture was stirred at room temperature for 4 h. Next, the mixture was poured into crushed ice water to obtain the precipitate. The precipitate was then filtered and washed by water. In the case of there being no formation of the precipitate, the solution would be extracted by EtOAc (3 × 15.0 mL). After that, the organic layers were combined and washed by water (2 × 15.0 mL). The organic layer was then dried by anhydrous Na_2_SO_4_ and evaporated under reduced pressure to obtain the crude product. Finally, the product was purified by preparative TLC or column chromatography (silica gel, hexane:EtOAc (4:1)).

##### (3*Z*)-1-Acetyl-3-benzylidene-4-methyl-2,5-diketopiperazine (**12a**)

Following the general procedure, the product was obtained as a white solid (0.03 g, 23% yield); m.p. 160–161 °C; ^1^H-NMR (300 MHz, CDCl_3_) δ 7.42–7.37 (m, 3H), 7.34–7.31 (m, 2H), 4.54 (s, 2H), 2.91 (s, 3H), 2.64 (s, 3H); ^13^C-NMR (300 MHz, CDCl_3_) δ 171.5, 164.9, 163.8, 132.7, 131.6, 129.6, 129.4, 128.6, 125.5, 45.3, 34.4, 26.7; HRMS [ESI]^+^ calculated for C_14_H_14_N_2_O_3:_ 281.0897 [M + Na]^+^; found: 281.0896.

##### (3*Z*)-1-Acetyl-4-allyl-3-benzylidene-2,5-diketopiperazine (**12b**)

Following the general procedure, the product was obtained as a white solid (0.02 g, 15% yield); m.p. 142–144 °C; ^1^H-NMR (300 MHz, CDCl_3_) δ 7.44–7.34 (m, 5H), 7.31 (s, 1H), 5.57–5.46 (m, 1H), 5.03 (dd, *J* = 10.2, 1.2 Hz, 1H), 4.74 (dd, *J* = 16.8, 1.2 Hz, 1H), 4.54 (s, 2H), 4.11 (d, *J* = 5.6 Hz, 2H), 2.63 (s, 3H); ^13^C-NMR (300 MHz, CDCl_3_) δ 171.3, 164.8, 164.3, 132.6, 131.0, 129.7, 129.6, 129.3, 128.7, 126.6, 118.8, 46.4, 45.2, 26.6; HRMS [ESI]^+^ calculated for C_16_H_16_N_2_O_3:_ 307.1053 [M + Na]^+^; found: 307.1050.

##### (3*Z*)-1-Acetyl-3-benzylidene-4-(3-methylbut-2-ene-1-yl)-2,5-diketopiperazine (**12c**)

Following the general procedure, the product was obtained as a colorless oil (0.11 g, 14% yield); ^1^H-NMR (300 MHz, CDCl_3_) δ 7.43–7.36 (m, 5H), 7.29 (s, 1H), 4.87 (tq, J = 6.9, 1.2 Hz, 1H), 4.50 (s, 2H), 4.11 (d, J = 6.9 Hz, 2H), 2.62 (s, 3H), 1.57 (s, 3H), 1.20 (s, 3H); ^13^C-NMR (300 MHz, CDCl_3_) δ 171.4, 164.7, 164.4, 132.5, 131.2, 129.6, 129.5, 129.3, 128.4, 121.5, 118., 46.0, 45.1, 26.6; HRMS [ESI]^+^ calculated for C_18_H_20_N_2_O_3:_ 335.1366 [M + Na]^+^; found: 335.1352.

##### (6*Z*)-4-Acetyl-6-(2-methylpropylidene)-1-methyl-2,5-diketopiperazine (**12d**)

Following the general procedure, the product was obtained as a brown oil (0.08 g, 14% yield); ^1^H-NMR (300 MHz, CDCl_3_) δ 6.17 (d, *J* = 11.1 Hz, 1H), 4.39 (s, 2H), 3.26 (s, 3H), 2.84–2.72 (m, 1H), 2.56 (s, 3H), 1.13 (d, *J* = 6.3 Hz, 6H); ^13^C-NMR (300 MHz, CDCl_3_) δ 171.4, 164.7, 164.0, 137.0, 131.1, 45.3, 34.8, 27.4, 26.6, 22.3; HRMS [ESI]^+^ calculated for C_11_H_16_N_2_O_3:_ 247.1053 [M + Na]^+^; found: 247.1044.

##### (6*Z*)-4-Acetyl-1-allyl-6-(2-methylpropylidene)-2,5-diketopiperazine (**12e**)

Following the general procedure, the product was obtained as a yellow oil (0.10 g, 18% yield); ^1^H-NMR (300 MHz, CDCl_3_) δ 6.17 (d, *J* = 10.8 Hz, 1H), 5.88–5.75 (m, 1H), 5.18 (dq, *J* = 18.9, 1.2 Hz, 2 H), 4.38 (s, 2H), 4.28 (at, *J* = 5.7, 1.5 Hz, 2H), 2.77–2.69 (m, 1H), 2.56 (s, 3H), 1.08 (d, *J* = 6.6 Hz, 6H); ^13^C-NMR (300 MHz, CDCl_3_) δ 170.3, 163.6, 163.4, 137.0, 130.9, 128.9, 116.7, 48.1, 44.3, 26.2, 25.5, 20.9; HRMS [ESI]^+^ calculated for C_13_H_18_N_2_O_3:_ 273.1210 [M + Na]^+^; found: 273.1207.

#### 3.2.9. (3*Z*)-1-Acetyl-3-benzylidene-2,5-diketopiperazine (**12f**)

1,4-Diacetyl piperazine-2,5-dione (**11**) (0.50 g, 2.5 mmol) was dissolved in dry DMF (10.0 mL) under argon atmosphere. Next, Cs_2_CO_3_ (0.80 g, 2.5 mmol) and benzaldehyde (0.2 mL, 1.8 mmol) were added into the solution. The mixture was then stirred at room temperature for 3 h. After the reaction was completed, the mixture was poured into crush ice water. (3*Z*)-1-Finally, Acetyl-3-benzylidene-2,5-diketopiperazine was precipitated as a white solid, which was filtered and washed by excess water (0.28 g, 46% yield); m.p. 185–189 °C (Lit [38] 195–197 °C) ^1^H-NMR (300 MHz, CDCl_3_) δ 7.93 (br s, 1H), 7.50–7.45 (m, 2H), 7.41–7.38 (m, 3H), 7.19 (s, 1H), 4.52 (s, 2H), 2.66 (s, 3H); ^13^C-NMR (300 MHz, CDCl_3_) δ 172.5, 162.7, 159.9, 132.5, 1129.6, 129.4, 128.5, 125.7, 119.9, 46.1, 27.2.

#### 3.2.10. (3*Z*)-1-Acetyl-4-benzyl-3-benzylidene-2,5-diketopiperazine (**12g**)

(3*Z*)-1-Acetyl-3-benzylidene-2,5-diketopiperazine (0.10 g, 0.4 mmol) was treated by K_2_CO_3_ (0.11 g, 0.8 mmol) with dry DMF (3.0 mL). Next, Benzyl chloride (0.06 g, 0.5 mmol) was added to the solution. The mixture was then stirred at room temperature overnight. After the reaction was completed, the reaction mixture was poured into water and extracted by EtOAc (3 × 15.0 mL). After that, the organic layer was washed with water and dried with anhydrous Na_2_SO_4._ Finally, the organic solvent was removed under reduced pressure to provide the crude product. (3*Z*)-1-Acetyl-4-benzyl-3-benzylidene-2,5-diketopiperazine (**18**) was then obtained as a pale yellow solid by column chromatography using Hexane:EtOAc (4:1) as an eluent (0.02 g, 17% yield); m.p. 117–120 °C; ^1^H-NMR (300 MHz, CDCl_3_) δ 7.50–7.39 (m, 5H), 7.21–7.19 (m, 3H), 6.88–6.85 (m, 2H), 4.67 (s, 2H), 4.55 (s, 2H), 2.54 (s, 3H); ^13^C-NMR (300 MHz, CDCl_3_) δ 171.3, 165.1, 164.3, 135.8, 132.6, 129.9, 129.7, 129.6, 128.9, 128.6, 127.9, 127.8, 126.9, 47.2, 45.2, 26.5; HRMS [ESI]^+^ calculated for C_20_H_18_N_2_O_3:_ 357.1210 [M + Na]^+^; found: 357.1205.

#### 3.2.11. General Procedure for the Synthesis of (3*Z*,6*Z*)-3-Benzylidene-6-(2-methyl propylidene)-4-substituted-2,5-diketopiperazine (**13a**–**f**)

The solution of intermediate (**12a**–**f**) (0.3 mmol, 1.0 eq) in dry DMF (3.0 mL) was treated by Cs_2_CO_3_ (0.4 mmol, 1.5 eq) and aldehyde (0.4 mmol, 1.5 eq) under argon atmosphere. The reaction mixture was stirred at room temperature for 3 h. Next, the mixture was poured into crushed ice water. After the precipitate was formed, it was filtered and washed by water. In case there was not formation of the precipitate, the solution would be extracted by EtOAc (3 × 15.0 mL). The organic layers were then combined and washed by water (2 × 15.0 mL). After that, the organic layer was dried with anhydrous Na_2_SO_4_ and evaporated under reduced pressure to obtain the crude product. Finally, the product was purified by preparative TLC or column chromatography (silica gel, hexane:EtOAc (4:1)).

##### (3*Z*,6*Z*)-3-Benzylidene-6-(2-methylpropylidene)-4-methyl-2,5-diketopiperazine (**13a**)

Following the general procedure, the product was provided as a pale yellow solid (0.06 g, 55% yield); m.p. 139–142 °C; ^1^H-NMR (300 MHz, CDCl_3_) δ 8.77 (br s, 1H), 7.43–7.31 (m, 3H), 7.28–7.25 (m, 3H), 6.05 (d, *J* = 10.2 Hz, 1H), 2.94 (s, 3H), 2.75–2.63 (m, 1H), 1.12 (d, *J* = 6.6 Hz, 6H); ^13^C-NMR (300 MHz, CDCl_3_) δ 160.2, 159.7, 134.0, 130.4, 129.4, 128.4, 128.2, 128.0, 124.5, 120.5, 36.5, 25.5, 22.1; HRMS [ESI]^+^ calculated for C_16_H_18_N_2_O_2:_ 293.1260 [M + Na]^+^; found: 293.1256.

##### (3*Z*,6*Z*)-4-Allyl-3-benzylidene-6-(2-methylpropylidene)-2,5-diketopiperazine (**13b**)

Following the general procedure, the product was obtained as a colorless oil (0.05 g, 46% yield); ^1^H-NMR (300 MHz, CDCl_3_) δ 9.53 (br s, 1H), 7.41–7.27 (m, 5H), 7.23 (s, 1H), 6.08 (d, *J* = 10.2 Hz, 1H), 5.59–5.45 (m, 1H), 5.00 (dd, *J* = 10.2, 1.2 Hz. 1H), 4.74 (dd, *J* = 17.1, 1.2 Hz, 1H), 4.23 (d, *J* = 5.7 Hz, 2H), 2.90–2.74 (m, 1H), 1.13 (d, *J* = 6.6 Hz, 6H); ^13^C-NMR (300 MHz, CDCl_3_) δ 161.4, 159.9, 134.0, 131.6, 129.3, 129.1, 128.7, 128.4, 124.7, 121.5, 117.9, 47.7, 25.5, 22.2; HRMS [ESI]^+^ calculated for C_18_H_20_N_2_O_2:_ 319.1417 [M + Na]^+^; found: 319.1410.

##### (3*Z*,6*Z*)-3-Benzylidene-6-(2-methylpropylidene)-4-(3-methyl but-2-en-1-yl)-2,5-diketopiperazine (**13c**)

Following the general procedure, the product was obtained as a white solid (0.05 g, 45% yield); m.p. 175–178 °C; ^1^H-NMR (300 MHz, CDCl_3_) δ 7.83 (br s, 1H), 7.40–7.27 (m, 5H), 7.23 (s, 1H), 6.03 (d, *J* = 10.2 Hz, 1H), 4.91 (tt, *J* = 6.6, 1.5 Hz, 1H), 4.20 (d, *J* = 6.6 Hz, 2H), 2.59–2.52 (m, 1H), 1.56 (s, 3H), 1.21 (s, 3H), 1.12 (d, *J* = 6.6 Hz, 6H); ^13^C-NMR (300 MHz, CDCl_3_) δ 161.1, 160.0, 136.9, 134.0, 129.4, 128.6, 128.5, 128.3, 128.1, 124.8, 121.6, 118.6, 44.0, 25.6, 25.5, 22.1, 17.7; HRMS [ESI]^+^ calculated for C_20_H_24_N_2_O_2:_ 347.1730 [M + Na]^+^; found: 347.1723.

##### (3*Z*,6*Z*)-3-Benzylidene-6-(2-methylpropylidene)-1-methyl-2,5-diketopiperazine (**13d**)

Following the general procedure, the product was obtained as a colorless oil (0.006 g, 8% yield); ^1^H-NMR (300 MHz, CDCl_3_) δ 8.05 (br s, 1H), 7.45–7.30 (m, 5H), 7.00 (s, 1H), 6.07 (d, *J* = 11.1 Hz, 1H), 3.43 (s, 3H), 3.01–2.89 (m, 1H), 1.13 (d, *J* = 6.5 Hz, 6H); ^13^C-NMR (300 MHz, CDCl_3_) δ 159.6, 159.0, 133.0, 131.7, 129.3, 128.6, 128.5, 128.4, 125.9, 116.5, 35.5, 27.4, 23.0; HRMS [ESI]^+^ calculated for C_16_H_18_N_2_O_2:_ 293.1260 [M + Na]^+^; found: 293.1251.

##### (3*Z*,6*Z*)-1-Allyl-3-benzylidene-6-(2-methylpropylidene)-2,5-diketopiperazine (**13e**)

Following the general procedure, the product was obtained as a pale yellow solid (0.02 g, 21% yield); m.p. 115–119 °C; ^1^H-NMR (300 MHz, CDCl_3_) δ 7.95 (br s, 1H), 7.46–7.34 (m, 5H), 7.01 (s, 1H), 6.13 (d, *J* = 11.4 Hz, 1H), 5.99–5.89 (m, 1H), 5.24 (ddd, *J* = 19.8, 10.5, 0.9 Hz, 2H), 4.49 (dt, *J* = 4.5, 1.8 Hz, 2H), 2.93–2.89 (m, 1H), 1.10 (d, *J* = 6.6 Hz, 6H); ^13^C-NMR (300 MHz, CDCl_3_) δ 159.4, 158.5, 133.1, 132.2, 131.7, 129.4, 128.7, 128.4, 127.0, 125.8, 116.5, 116.3, 49.2, 27.0, 22.9; HRMS [ESI]^+^ calculated for C_18_H_20_N_2_O_2:_ 319.1417 [M + Na]^+^; found: 319.1420.

##### (3*Z*,6*Z*)-3-Benzylidene-6-(2-methylpropylidene)-2,5-diketopiperazine (**13f**)

Following the general procedure, the product was obtained as a white solid (0.13 g, 82% yield); m.p. 243–246 °C (Lit [39] 271–271.5 °C); ^1^H-NMR (300 MHz, CDCl_3_) δ 8.06 (br s, 1H), 7.81 (br s, 1H), 7.48–7.43 (m, 2H), 7.38–7.34 (m, 3H), 6.99 (s, 1H), 6.02 (d, *J* = 10.2 Hz, 1H), 2.59–2.51 (m, 1H), 1.12 (d, *J* = 6.6 Hz, 6H); ^13^C-NMR (300 MHz, CDCl_3_) δ 157.1, 157.0, 132.7, 129.5, 128.9, 128.3, 126.8, 124.3, 116.2, 29.7, 25.5, 22.0.

##### (3*Z*,6*Z*)-4-Benzyl-3-benzylidene-6-(2-methylpropylidene)-2,5-diketopiperazine (**13g**)

Following the general procedure, the product was obtained as a colorless oil (0.01 g, 12% yield); ^1^H-NMR (300 MHz, CDCl_3_) δ 7.95 (br s, 1H), 7.44–7.32 (m, 4H), 7.30–7.27 (m, 1H), 7.21 (s, 1H), 7.19–7.16 (m, 3H), 6.92–6.83 (m, 2H), 6.05 (d, *J* = 10.5 Hz, 1H), 4.72 (s, 2H), 2.60–2.50 (m, 1H), 1.11 (d, *J* = 6.6 Hz, 6H); ^13^C-NMR (300 MHz, CDCl_3_) δ 160.6, 160.0, 136.2, 133.9, 129.5, 128.8, 128.6, 128.5, 128.4, 128.3, 127.5, 124.6, 122.0, 48.6, 29.7, 25.7, 22.1; HRMS [ESI]^+^ calculated for C_22_H_22_N_2_O_2:_ 369.1573 [M + Na]^+^; found: 369.1567.

#### 3.2.12. General Procedure for the Synthesis of (3*Z*,6*Z*)-3-Benzylidene-6-(2-methylpropylidene)-1,4-disubstituted-2,5-diketopiperazine (**14a**–**b**)

(3*Z*,6*Z*)-3-Benzylidene-6-(2-methylpropylidene)-2,5-diketopiperazine (**13f**) ((0.08 g, 0.30 mmol) was dissolved in DMF (3.0 mL) under argon atmosphere. Next, Cs_2_CO_3_ (0.32 g, 0.90 mmol) and alkyl halide (0.06 mL, 0.90 mmol) were added to a solution. The reaction mixture was then stirred at 0 °C for 2 h. The reaction was quenched with brine and extracted by EtOAc (2 × 20.0 mL). After that, the organic layer was washed by brine (3 × 20.0 mL) and dried with anhydrous Na_2_SO_4._ The crude product was obtained by removing EtOAc solvent under reduced pressure. Finally, the product was purified by preparative TLC, using methanol in DCM (5%) as a mobile phase.

##### (3*Z*,6*Z*)-3-Benzylidene-6-(2-methylpropylidene)-1,4-bis(-3-methylbut-2-en-1yl)-2,5-diketopiperazine (**14a**)

Following the general procedure, the product was obtained as a colorless oil (0.002 g, 2% yield); ^1^H-NMR (300 MHz, CDCl_3_) δ 7.36–7.29 (m, 5H), 7.09 (s, 1H), 6.01 (d, J = 10.8 Hz, 1H), 5.23 (tq, *J* = 6.1, 1.5 Hz, 1H), 4.88 (tq, *J* = 6.8, 1.4 Hz, 1H), 4.35 (d, *J* = 6.1 Hz, 2H), 4.09 (d, *J* = 6.8 Hz, 2H), 2.80–2.68 (m, 1H), 1.71 (sd, *J* = 1.1 Hz, 3H), 1.68 (s, 3H), 1.55 (sd, *J* = 0.8 Hz, 3H), 1.25 (sd, *J* = 0.6 Hz, 3H), 1.09 (d, *J* = 6.5 Hz, 6H); ^13^C-NMR (300 MHz, CDCl_3_) δ 163.4, 162.9, 136.9, 136.0, 129.9, 129.5, 129.4, 128.6, 128.3, 122.2, 119.5, 118.7, 45.5, 43.3, 27.1, 25.6, 25.5, 22.5, 18.2, 17.6; HRMS [ESI]^+^ calculated for C_25_H_32_N_2_O_2:_ 393.2537 [M + H]^+^; found: 393.2542.

##### (3*Z*,6*Z*)-1,4-Dibenzyl-3-benzylidene-6-(2-methylpropylidene)-2,5-diketo piperazine (**14b**)

Following the general procedure, the product was obtained as a pale yellow oil (0.02 g, 10% yield); ^1^H-NMR (300 MHz, CDCl_3_) δ 7.44–7.27 (m, 5H), 7.25 (s, 1H), 7.23–7.18 (m, 3H), 7.16–7.10 (m, 3H), 7.08–7.01 (m, 2H), 6.79–6.76 (m, 2H), 6.04 (d, *J* = 10.8 Hz, 1H), 4.94 (s, 2H), 4.63 (s, 2H), 2.79–2.67 (m, 1H), 1.07 (d, *J* = 6.6 Hz, 6H); ^13^C-NMR (300 MHz, CDCl_3_) δ 163.5, 163.0, 136.4, 134.4, 133.4, 129.9, 129.6, 129.3, 129.0, 128.6, 128.4, 127.7, 127.6, 127.5, 126.6, 122.9, 50.2, 48.3, 27.3, 22.2; HRMS [ESI]^+^ calculated for C_29_H_28_N_2_O_2:_ 459.2043 [M + Na]^+^; found: 459.2052.

##### (3*Z*,6*E*)-3-Benzylidene-6-(2-methylpropylidene)-1-(1-ethyl pyrrolidine)-2,5-diketopiperazine (**14c**)

Following the general procedure, the product was obtained as a pale yellow solid (0.01 g, 6% yield); m.p. 158–162 °C; ^1^H-NMR (300 MHz, CDCl_3_) δ 8.16 (br s, 1H), 8.05 (d, *J* = 7.2 Hz, 2H), 7.40–7.28 (m, 3H), 7.27 (s, 1H), 5.56 (d, *J* = 10.2 Hz, 1H), 4.58 (t, *J* = 6.0 Hz, 2H), 3.04 (t, *J* = 6.0 Hz, 2H), 2.76 (m, 4H), 2.66–2.58 (m, 1H), 1.88 (sq, *J* = 3.3 Hz, 4H), 1.10 (d, *J* = 6.6 Hz, 6H); ^13^C-NMR (300 MHz, CDCl_3_) δ 160.4, 153.6, 135.1, 131.5, 128.7, 128.3, 127.8, 123.0, 121.2, 121.1, 65.6, 54.6, 54.1, 25.0, 23.5, 22.3; HRMS [ESI]^+^ calculated for C_21_H_27_N_2_O_2:_ 354.2103 [M + H]^+^; found: 354.2174.

### 3.3. Virus Propagation Inhibition Assay

Virus propagation inhibition assays were examined by embryonated chicken egg inoculation. One hundred microliters of LS-C, LS-D, and 2,5-DKP derivatives at various concentrations (25, 50, and 100 µg mL^−1^) in 0.1 M PBS (pH 7.2) (from DMSO stock) were incubated with 100 µL of virus suspension at 37 °C for 30 min. The mixture (100 µL) was inoculated into each embryonated chicken egg and incubated at 37 °C for 4 days. Virus dissolved in saline solution was used as a negative control. The allantoic fluid was investigated by Hemagglutination assay [40].

### 3.4. Hemagglutination Assay (HA)

Fresh chicken blood was taken in Alsever’s solution and centrifuged at 2000 rpm for 5 min. The chicken red blood cells (RBCs) were separated from the supernatant and washed with PBS until the supernatant was cleared. The 2.5% suspension of RBCs was prepared by mixing with PBS. Later, the allantoic fluid 100 µL, which was harvested from inoculated embryonate chicken eggs, was mixed with PBS 50 µL and diluted with 2-fold serial dilution in PBS from 1:1 to 1:3072 in a 96-well micro titer plate. 1-Adamantanamine hydrochloride (Sigma-Aldrich, St. Louis, MO, USA) used a positive control, and PBS used as negative control, were included on micro plates. The 2.5% RBCs suspension 100 µL was added and mixed to every well. The microplate was kept at 4 °C for 45–60 min. Negative results displayed of the RBCs precipitation by gravity to the bottom of the well demonstrated absence of the virus, while positive results displayed the formation of a diffuse mat on the bottom of the well, leading to a lattice formation of RBCs and the virus. The HA titers were reported as the end point of the virus’s last dilution, showing complete agglutination. The experiments were performed in triplicate.

## 4. Cytotoxicity

Cytotoxicity was evaluated against normal cell line Rhesus monkey kidney epithelial cells LLC-MK2 (Korean Cell Line Bank, KCLB) by using a MTT assay [41]. A stock solution of LS-C, LS-D, and 2,5-DKP derivatives was prepared at the concentration of 5000 µg/mL in EtOH. The stock solution was further diluted to 128 µg/mL prior to use and added to the first wells. Two-fold serial dilutions were employed to obatin a by culture medium with final concentration of 1 µg/mL. Cells were seeded at a density of 5 × 10^4^ cells/well and incubated for 16 h in a 96 well plate, followed by the treatment with the test compounds. 2.5% DMSO was used as a control culture. After 24 h, LLC-MK2 cells were incubated with MTT (500 μg/mL) for 4 h. DMSO was then added to dissolve the formation of blue formazan crystals. Finally, optical density at 450 nm was determined by a microplate reader.

## 5. Molecular Docking Study

Molecular docking study was performed using iGEMDOCK (Generic Evolutionary Method for Molecular DOCKing) v.2.1 (Institute of Bioinformatics National Chiao-Tung University, Hsinchu, Taiwan) to investigate the possible binding between neuraminidase from H5N2 avian influenza virus (H5N2, PDB ID: 5HUK) and our compounds (LS-C, compound **7**, **9**, **13a**–**13d**, and **14c**) to explore the capabilities of the compounds as virus propagation inhibitors in comparison to oseltamivir carboxylate, zanamivir, and peramivir. The accurate docking (very slow) with population size (N = 800), 80 generation, and 10 solutions were applied for the docking of each ligand against protein. The docking pose with the lowest binding energy value for each ligand–protein complex was then analyzed and imaged using BIOVIA Discovery Studio Visualizer [24].

## 6. Conclusions

2,5-Diketopiperazine derivatives have been successfully prepared by addition of benzylidene and isopropylidene substituents based on the structure of Albonoursin. Using hemagglutination assay, their antiviral activity against influenza virus (H5N2) was investigated and compared with those of the natural compounds with a similar structure (Lansai C and D). The results showed that LS-C and compounds **13b**–**d** and **14c** exhibited antiviral activity against influenza virus (H5N2) at a concentration of 25 µg/mL. Furthermore, the molecular docking study showed that these compounds could bind and strongly interact to key amino acid residues in 430-cavity of neuraminidase from H5N2 avian influenza virus.

## Data Availability

The data presented in this study are available on request from the corresponding authors.

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
