# Peer review of "2,5-Diketopiperazine Derivatives as Potential Anti-Influenza (H5N2) Agents: Synthesis, Biological Evaluation, and Molecular Docking Study"

_molecules, 2022, doi:10.3390/molecules27134200_

Round 1

Reviewer 1 Report

The manuscript by Winyakul and colleagues presents the synthesis of a number of diketopiperazine derivatives that have been experimental investigated against influenza virus and computationally against SARS-CoV-2. The data are presented in a very confusing way and the also the English language does not help clarity of presentation. The authors started from natural compound Lansai C and D that are initially described as reference structures to be tested against influenza virus, the rationale of the synthesized compound is poorly described and it is not clear what the Authors wish to investigate. Suprisingly, the compounds synthesized are opposite regioisomers of Lansai C and D and analogues of another natural compounds class (Albonoursin) which actually has anti-influenza activity. Then the Authors attempts to correlate structure to activity, which is not an easy task for compounds investigated phenotypically. The Authors seems not to be highly expert in the field, since they try to compare N-OH with N-alkyl substitutions, or they define allyl or butenyl chains as “electrostatic groups”. Regarding to anti-H5N2 investigation, neither in the main text, neither in the experimental part it is explained how the results are presented: what is the meaning of 1:1.5, 1:12 and so on? Also the docking part has serious problem, one of them being favipiravir docked in the four enzymes investigated without any rationale

For these reasons, despite the synthetic efforts, I think this paper cannot be published in the current form

Author Response

            This is our response related to the manuscript entitled, “2,5-Diketopiperazine derivatives as potential antiviral agents: synthesis, biological evaluations, and anti-SAR-CoV2 activity studies using molecular docking” submitted to Molecules. In response to all of the comments by the reviewer, we have already performed further revised the presentation points in the revised version as labeled in blue (in the attached file).  

Reviewer #1:

The manuscript by Winyakul and colleagues presents the synthesis of a number of diketopiperazine derivatives that have been experimental investigated against influenza virus and computationally against SARS-CoV-2.

Comment No 1.  The data are presented in a very confusing way and the also the English language does not help clarity of presentation.

Response to the Comment No 1: We thanks the reviewer for carefully reading our manuscript. As suggested, where it is necessary for clarity, we have improved the English language of the manuscript throughout.

Comment No 2. The authors started from natural compound Lansai C and D that are initially described as reference structures to be tested against influenza virus, the rationale of the synthesized compound is poorly described and it is not clear what the Authors wish to investigate. Suprisingly, the compounds synthesized are opposite regioisomers of Lansai C and D and analogues of another natural compounds class (Albonoursin) which actually has anti-influenza activity. Then the Authors attempts to correlate structure to activity, which is not an easy task for compounds investigated phenotypically. The Authors seems not to be highly expert in the field, since they try to compare N-OH with N-alkyl substitutions, or they define allyl or butenyl chains as “electrostatic groups”.

Response to the Comment No 2: We thanks the reviewer for comments. The work was focused on the synthesis of N-substituted Albonoursin and their antiviral activity. The natural compound Lansai C and D, which had similar structure with Albonoursin except the double bond configuration, were also tested for their antiviral activity. Therefore, as suggested, we have revised and modified the introduction part to give clearer main idea of this work as shown in blue in the manuscript. Moreover, we have modified the results of the biological activity part. The activities of N-substituted Albonoursin and Lansai C and D were separately discussed as shown in blue in the manuscript.

Comment No 3. Regarding to anti-H5N2 investigation, neither in the main text, neither in the experimental part it is explained how the results are presented: what is the meaning of 1:1.5, 1:12 and so on?

Response to the Comment No 3: We thanks the reviewer for pointing this out. As suggested, the virus propagation inhibition assay and emagglutination assay have been added in the experimental section. The results were explained in more details and the meaning of the ratio was added underneath the table as shown in blue in the manuscript.

Comment No 4. Also the docking part has serious problem, one of them being docked in the four enzymes investigated without any rationale.

Response to the Comment No 4: We thanks the reviewer for this comment. We agree with the reviewer and have revised and modified the docking part as shown in blue in the manuscript.

We are grateful to the reviewers for their kind comments and suggestions.

Reviewer 2 Report

  1. Preparation of X-ray Single crystal structure for one of the new products could be promote the quality of the paper.
  2. Please compare the influenza virus (H5N2) propagation inhibition activity of the synthesized compounds quantitatively with a standard and reference compound.
  3. In molecular docking study section: The maximum value for binding energy from previous studies is around -20 kcal/mol [Smith, Richard D., et al. "Biophysical limits of protein–ligand binding." Journal of chemical information and modeling 52 (2012): 2098-2106]. How the authors came up with these numbers is unknown. Numbers such as -91.93(13c and Avian influenza virus), -77.87(13b and Avian influenza virus), and -101.86 (for interaction between 13d and Avian influenza virus) for binding energy are incorrect and should be rewritten. These are just three examples for authors. Every docking result needs to be adjusted.

Molecules with too high of a binding affinity can exhibit clearance problems in the body [. Kuntz ID, Chen K, Sharp KA, Kollman PA. The maximal affinity of ligands. Proc. Natl. Acad. Sci. U. S. A. 1999; 96:9997–10002].

  1. In addition to other outcomes of binding data, they should also provide the RMSD values.
  2. According to the conclusion and aim of the article, pharmacophore and QSAR 2D and 3D are necessary for this kind of study to support the conclusions.
  3. Poor grammar and vocabulary structure. Consider using Grammarly and other options.

Author Response

            This is our response related to the manuscript entitled, “2,5-Diketopiperazine derivatives as potential antiviral agents: synthesis, biological evaluations, and anti-SAR-CoV2 activity studies using molecular docking” submitted to Molecules. In response to all of the comments by the reviewer, we have already performed further revised the presentation points in the revised version as labeled in blue (in the attached file).  

Reviewer #2:

Comment No 1. Preparation of X-ray Single crystal structure for one of the new products could be promote the quality of the paper.

Response to the Comment No 1: We thanks the reviewer for this suggestion. We have tried many attempts to recrystallize our solid products however they were unsuccessful to obtain the single crystals to run X-ray crystallography.

Comment No 2. Please compare the influenza virus (H5N2) propagation inhibition activity of the synthesized compounds quantitatively with a standard and reference compound.

Response to the Comment No 2: We thanks the reviewer for raising an important point. We have added the positive control as 1-Adamantanamine hydrochloride: a standard drug used in the prophylactic or symptomatic treatment of influenza A.

Comment No 3. In molecular docking study section: The maximum value for binding energy from previous studies is around -20 kcal/mol [Smith, Richard D., et al. "Biophysical limits of protein–ligand binding." Journal of chemical information and modeling 52 (2012): 2098-2106]. How the authors came up with these numbers is unknown. Numbers such as -91.93(13c and Avian influenza virus), -77.87(13b and Avian influenza virus), and -101.86 (for interaction between 13d and Avian influenza virus) for binding energy are incorrect and should be rewritten. These are just three examples for authors. Every docking result needs to be adjusted.

Molecules with too high of a binding affinity can exhibit clearance problems in the body [. Kuntz ID, Chen K, Sharp KA, Kollman PA. The maximal affinity of ligands. Proc. Natl. Acad. Sci. U. S. A. 1999; 96:9997–10002].

Response to the Comment No 3: We thanks the reviewer for pointing this out. The binding energies demonstrated in the manuscript were from molecular docking results, predicted using iGEMDOCK v 2.1 software. The binding energy of the same system can be different if different molecular docking software is used. Therefore, to judge the binding ability of drug candidate, the binding energies of drug candidates are normally compared among themselves or compared with that of commercial drug, calculated using the same software [1-3]. For our results, we selected Oseltamivir and Favipiravir as approved antivirus drugs and their molecular docking results were then used to compare with those our 2,5-diketopiperazine derivatives in investigate their possible anti-influenza medications and anti-SAR-CoV2 activity, respectively.

  1. Abdusalam, A. A. A. and Murugaiyah, V. (2020). Identification of potential inhibitors of 3CL protease of SARS-CoV-2 from ZINC database by molecular docking-based virtual screening. Frontiers in molecular biosciences, 7, 419.
  2. Palakhachane, S., et al. (2021). Synthesis of sorafenib analogues incorporating a 1, 2, 3-triazole ring and cytotoxicity towards hepatocellular carcinoma cell lines. Bioorganic Chemistry, 112, 104831.
  3. Hasan, M. K., et al. (2021). Structural analogues of existing anti-viral drugs inhibit SARS-CoV-2 RNA dependent RNA polymerase: A computational hierarchical investigation. Heliyon, 7(3), e06435.

Comment No 4. In addition to other outcomes of binding data, they should also provide the RMSD values.

Response to the Comment No 4: We thanks the reviewer for this comment. RMSD values are normally provided for molecular dynamics simulations to indicates simulation equilibration and its fluctuation around a thermal mean. For molecular docking studies, RMSD values are generally not provided as can be seen the previous works [4-8].

  1. Shaikh, I. A., et al. (2022). In Silico Molecular Docking and Simulation Studies of Protein HBx Involved in the Pathogenesis of Hepatitis B Virus-HBV. Molecules, 27(5), 1513.
  2. Vishvakarma, V. K., et al. (2022). Hunting the main protease of SARS-CoV-2 by plitidepsin: Molecular docking and temperature-dependent molecular dynamics simulations. Amino acids, 54(2), 205-213.
  3. Adebambo, K. F., et al. (2021). Molecular Docking Study of the Binding Interaction of Hydroxychloroquine, Dexamethasone and Other Anti-Inflammatory Drugs with SARS-CoV-2 Protease and SARS-CoV-2 Spikes Glycoprotein. Computational Molecular Bioscience, 11(2), 19-49.
  4. Vincent, S., et al. (2020). Molecular docking studies on the anti-viral effects of compounds from kabasura kudineer on SARS-CoV-2 3CLpro. Frontiers in molecular biosciences, 7, 434.
  5. Attique, S. A., et al. (2019). A molecular docking approach to evaluate the pharmacological properties of natural and synthetic treatment candidates for use against hypertension. International journal of environmental research and public health, 16(6), 923.

Comment No 5. According to the conclusion and aim of the article, pharmacophore and QSAR 2D and 3D are necessary for this kind of study to support the conclusions.

Response to the Comment No 5: We thanks the reviewer for this suggestion. It would have been interesting to explore this aspect and it will be our future work. We have also modified the conclusion as shown below in blue and in manuscript.

“2,5-Diketopiperazine derivatives have been successfully prepared by adding benzylidene and isopropylidene substituents based on the structure of Albonoursin. Using hemagglutination assay, their antiviral activity against influenza virus (H5N2) were investigated and compared with that of the natural compound Lansai C and D, which had similar structure with Albonoursin except the double bond configuration. The results showed that LS-C and compounds 13b–d and 14c showed antiviral activity against influenza virus (H5N2) at a concentration of 25 µg/mL. Furthermore, the molecular docking study results showed that LS-C and compound 13d can fit in the binding pocket of both SARS-CoV-2 3CLpro and SARS-CoV-2 RBD with ACE2 with similar binding energy to Favipiravir, indicating their anti-SARS-CoV-2 potential. Additionally, it was found that the 2,5-DKP scaffold of our compounds could be the key to bind and inhibit the enzyme involved in antiviral activities. Our results, hence, would be useful for the design and development of new promising antiviral drugs.”

Comment No 6. Poor grammar and vocabulary structure. Consider using Grammarly and other options.

Response to the Comment No 6: We thanks the reviewer for carefully reading our manuscript. As suggested, where it is necessary for clarity, we have improved the English language of the manuscript throughout.

We are grateful to the reviewers for their kind comments and suggestions.

Yours sincerely,

Assistant Professor Dr. Waya S. Phutdhawong

Department of Chemistry, Faculty of Science,

Silpakorn University, 73000, Thailand.

Reviewer 3 Report

The manuscript submitted for Molecules journal has as main topic piperazine derivatives as potential antiviral agents. The work is important in terms of subjects for both industry and academic purposes. Besides, a good aspect of this work is the presence of the detailed syntheses and fully characterization by organic chemistry methods of all compounds involved in this study. However, there are some issues that needs attention:

-I suggest the reorganization of Abstracts, in order to make more easily understandable- the use of 13b-d and 14 numbers for compounds in Abstract is difficult to link to an actual structure.

-lines 46-62 are not arranges in justify mode;

-Introduction has only 9 references, in a very important subject, that is also well covered in literature-therefore some more literature data is necessary to be added;

-why are amino acid written in the middle of sentences with capital letters? (like in line 87); similar for Hexane (line 231) and so on;

-I suggest also for Schemes to be bigger, to make use of the space on the left of the page;

-for molecular docking study, more details about the soft used are necessary;

-what was the reason for choosing those four enzymes in molecular docking studies? this must be better explained;

-what is the significance and the meaning of the hydrogen bond length compiled in Table 2?

-some references numbers are written in bold; DOI numbers should be added.

Based on these comments, the work can be accepted for publication after minor corrections.

Author Response

            This is our response related to the manuscript entitled, “2,5-Diketopiperazine derivatives as potential antiviral agents: synthesis, biological evaluations, and anti-SAR-CoV2 activity studies using molecular docking” submitted to Molecules. In response to all of the comments by the reviewer, we have already performed further revised the presentation points in the revised version as labeled in blue (in the attached file).  

Reviewer #3:

The manuscript submitted for Molecules journal has as main topic piperazine derivatives as potential antiviral agents. The work is important in terms of subjects for both industry and academic purposes. Besides, a good aspect of this work is the presence of the detailed syntheses and fully characterization by organic chemistry methods of all compounds involved in this study. However, there are some issues that needs attention:

Comment No 1.  I suggest the reorganization of Abstracts, in order to make more easily understandable- the use of 13b-d and 14 numbers for compounds in Abstract is difficult to link to an actual structure.

Response to the Comment No 1: We thanks the reviewer for your comments. The names of compounds mentioned in the abstract have been replaced for easier understanding.

Comment No 2.  lines 46-62 are not arranges in justify mode;

Response to the Comment No 2: As suggested, the paragraph in line 46-62 has been justified.

Comment No 3.  Introduction has only 9 references, in a very important subject, that is also well covered in literature-therefore some more literature data is necessary to be added;

Response to the Comment No 3: We thanks the reviewer for your suggestion. We have added five more references in the introduction. 

Comment No 4.  why are amino acid written in the middle of sentences with capital letters? (like in line 87); similar for Hexane (line 231) and so on;

Response to the Comment No 4: We thanks the reviewer for your comment. We have rechecked and corrected all of the capital letters in the middle of sentences.

Comment No 5. I suggest also for Schemes to be bigger, to make use of the space on the left of the page;

Response to the Comment No 5: We thanks the reviewer for your comment. We have adjusted the Schemes to be bigger.

Comment No 6. for molecular docking study, more details about the soft used are necessary;

Response to the Comment No 6: We thanks the reviewer for this suggestion. We have added more details about iGEMDOCK v.2.1 software shown below in the experimental section the manuscript.

“Molecular docking studies were performed using iGEMDOCK (Generic Evolutionary Method for Molecular DOCKing) v.2.1 to investigate the possible binding between our compounds (LS-C, compound 7, compound 9, compounds 13a–13d and compound 14c), which serve as virus propagation inhibitor and enzymes involved in antiviral activities including neuraminidase from H5N2 avian influenza virus (H5N2, PDB ID: 5HUK), SARS-CoV-2 3CL main protease (SARS-CoV-2 3CLpro, PDB ID: 6LU7), and SARS-CoV-2 spike receptor-binding domain bound with ACE2, (SARS-CoV-2 RBD with ACE2, PDB ID: 6M0J). The accurate docking (very slow) with population size (N=800), 80 generation and 10 solutions was applied for docking each ligand against protein. The docking pose with lowest binding energy value for each ligand-protein complexes was then analyzed and imaged using BIOVIA Discovery Studio Visualizer [23].”

Comment No 7. what was the reason for choosing those four enzymes in molecular docking studies? this must be better explained;

Response to the Comment No 7: We thanks the reviewer for this comment. We have added the reason for choosing enzymes in molecular docking as shown below in blue and in the manuscript.

“In this work, there were three enzymes involved in antiviral activities selected as receptor including neuraminidase from H5N2 avian influenza virus (H5N2, PDB ID: 5HUK), SARS-CoV-2 3CL main protease (SARS-CoV-2 3CLpro, PDB ID: 6LU7), and SARS-CoV-2 spike receptor-binding domain bound with ACE2, (SARS-CoV-2 RBD with ACE2, PDB ID: 6M0J). Neuraminidase from H5N2 avian influenza virus is known as the drug target for the prevention of the spread of influenza infection. [24] On the other hand, SARS-CoV-2 3CLpro and SARS-CoV-2 RBD with ACE2 are potential therapeutic target for treating COVID-19 since SARS-CoV-2 3CLpro involves in the replication of the virus and SARS-CoV-2 RBD with ACE2 involves the first step of viral infection [25-27]. These two enzymes were used as receptor in molecular docking to explore potential anti-SARS-CoV-2 drugs in several reports. [28-33] Hence, the binding of our compounds in the cativity of SARS-CoV-2 3CLpro and SARS-CoV-2 RBD with ACE2 could indicate their possible anti-SAR-CoV2 activity. Additionally, Oseltamivir, an antiviral medication used to treat and prevent influenza viruses [34,35] was docked into Neuraminidase from H5N2 avian influenza virus, whereas Favipiravir, an antiviral medication used to treat influenza [36] was docked into SARS-CoV-2 3CLpro and SARS-CoV-2 RBD with ACE2 and their docking result were compared with our potential antiviral drugs (LS-C and compounds 7, 9, 13a–13d, and 14c).”

Comment No 8. what is the significance and the meaning of the hydrogen bond length compiled in Table 2?

Response to the Comment No 8: We thanks the reviewer for pointing this out. The hydrogen bond length in Table 2 indicates the strength of hydrogen bond. According to the Brown and Blessing criteria [1-2] of relating to the donor-acceptor distances (D… A), the hydrogen bond is considered to be weak if the donor-acceptor distance is greater than 2.73 Å, while the hydrogen bond is considered to be strong if the donor-acceptor distance is less than 2.73 Å.

  1. Brown, I. D. (1976). On the geometry of O–H⋯ O hydrogen bonds. Acta Crystallographica Section A: Crystal Physics, Diffraction, Theoretical and General Crystallography, 32(1), 24-31.
  2. Blessing, R. H. (1986). Hydrogen bonding and thermal vibrations in crystalline phosphate salts of histidine and imidazole. Acta Crystallographica Section B: Structural Science, 42(6), 613-621.

Comment No 9. some references numbers are written in bold; DOI numbers should be added.

Response to the Comment No 9: We thanks the reviewer for this suggestion. We have corrected the references numbers and added the DOI numbers.

We are grateful to the reviewers for their kind comments and suggestions.

Yours sincerely,

Assistant Professor Dr. Waya S. Phutdhawong

Department of Chemistry, Faculty of Science,

Silpakorn University, 73000, Thailand.

Round 2

Reviewer 1 Report

With respect to the previous version, the work of Winyakul was improved with respect to the quality of presentation. The introduction and rationale, as well as the scope of the work is clearer along with the antiviral results. However, the English language is still not adequate and the Authors should accurately proofread the text, maybe with the help of a native English speaker.

The major issue of this work remains the molecular modeling part which is merely speculative, poorly described both in the main text and in the experimental part and does not add anything to the rest of the work. Accordingly, the novel compounds developed within the study have experimentally proven antiviral activity against influenza virus, not coronavirus, so it does not make sense to perform the docking studies on SARS-CoV-2 enzymes. The docking study is not validated by any experiment, including the rference compound that are known to target different targets with respect to those investigated. Regarding docking on neuroaminidase enzyme, again the experimental part does not give much details to understand how the study was performed, Figure 4 is useless since the binding mode of the compounds is not clearly visible, and is not possible to see the structure of the reference drug that was used. Oseltamivir is a prodrug so using this compound as positive reference in the docking study does not make sense. So I suggest to remove the docking part from the work

Other revisions that could improve the quality of the manuscript are the following:

1. for the compounds that halt the spread of the infection up to 25 uM, it would be interesting to see at which concentration below 25 uM is still possible to observe full inhibition of viral spread

2. since the antiviral assay used by the Authors does not account for the vitality of the viral particle, for one/two best performing compounds it would be interesting to peform a plaque assay in order to assess this aspect

Author Response

            This is our response related to the manuscript entitled, “2,5-Diketopiperazine derivatives as potential antiviral agents: synthesis, biological evaluation and molecular docking study” submitted to Molecules. In response to all of the comments by the reviewer, we have already performed further revised the presentation points in the revised version as labeled in blue (in the attached file).  

Reviewer #1:

With respect to the previous version, the work of Winyakul was improved with respect to the quality of presentation. The introduction and rationale, as well as the scope of the work is clearer along with the antiviral results.

Comment No 1.  The English language is still not adequate and the Authors should accurately proofread the text, maybe with the help of a native English speaker.

Response to the Comment No 1: We thanks the reviewer for carefully reading our manuscript. As suggested, where it is necessary for clarity, we have improved the English language of the manuscript throughout.

Comment No 2. The major issue of this work remains the molecular modeling part which is merely speculative, poorly described both in the main text and in the experimental part and does not add anything to the rest of the work. Accordingly, the novel compounds developed within the study have experimentally proven antiviral activity against influenza virus, not coronavirus, so it does not make sense to perform the docking studies on SARS-CoV-2 enzymes. The docking study is not validated by any experiment, including the rference compound that are known to target different targets with respect to those investigated. Regarding docking on neuroaminidase enzyme, again the experimental part does not give much details to understand how the study was performed, Figure 4 is useless since the binding mode of the compounds is not clearly visible, and is not possible to see the structure of the reference drug that was used. Oseltamivir is a prodrug so using this compound as positive reference in the docking study does not make sense. So I suggest to remove the docking part from the work

Response to the Comment No 2: We thanks the reviewer for comments. As suggested, we have significantly revised and modified molecular docking part. The molecular docking study involved SAR-CoV2 was removed and the title of manuscript was also changed to “2,5-Diketopiperazine derivatives as potential antiviral agents: synthesis, biological evaluation and molecular docking study”. For docking on neuraminidase from H5N2 avian influenza virus, we have performed molecular docking study of oseltamivir carboxylate, zanamivir and peramivir, and compared these results to those of our compounds. Moreover, the binding mode results were represented much clearer in Figure 4 and more details and discussion involved binding mode and interactions observed were added as shown in blue in the manuscript. We believe that these results can be valuable guide for the development of new 2,5-DKP derivatives as new promising antiviral drugs.  

Comment No 3.  For the compounds that halt the spread of the infection up to 25 uM, it would be interesting to see at which concentration below 25 uM is still possible to observe full inhibition of viral spread

Response to the Comment No 3: Response to the Comment No 3: We thanks the reviewer for pointing this out. We have performed further experiment to test antiviral activity of our compounds at the concentration of 12.5 µg/mL (base on two-fold dilution) and no viral inhibition was observed. Therefore, according to hemagglutination assay, the lowest concentration showing full inhibition of viral spread for our compounds was 25 µg/mL. The results for the concentration of 12.5 µg/mL were also added in Table 1 as shown in blue in the manuscript.

Comment No 4. Since the antiviral assay used by the Authors does not account for the vitality of the viral particle, for one/two best performing compounds it would be interesting to peform a plaque assay in order to assess this aspect

Response to the Comment No 4: Response to the Comment No 4: We thanks the reviewer for this suggestion. For hemagglutination assay, the ratio red blood cells and viral particles at the end point of the complete agglutination titer is 1:1. As a result, the number of viral particles is equivalent to the amount of red blood cells at the end point of the complete agglutination titer. For our antiviral activity study, we preliminarily accounted the vitality of the viral particles in embryo chicken egg inoculation via the number of red blood cells. A plaque assay will be our future work We are grateful to the reviewers for their kind comments and suggestions.

Yours sincerely,

Assistant Professor Dr. Waya S. Phutdhawong

Department of Chemistry, Faculty of Science,

Silpakorn University, 73000, Thailand.

Reviewer 2 Report

The manuscript has been sufficiently improved to warrant publication in Molecules.

Author Response

            This is our response related to the manuscript entitled, “2,5-Diketopiperazine derivatives as potential antiviral agents: synthesis, biological evaluation and molecular docking study” submitted to Molecules. In response to all of the comments by the reviewer, we have already performed further revised the presentation points in the revised version as labeled in blue (in the attached file).  

We are grateful to the reviewers for their kind comments and suggestions.

Yours sincerely,

Assistant Professor Dr. Waya S. Phutdhawong

Department of Chemistry, Faculty of Science,

Silpakorn University, 73000, Thailand.
